# A scientometric analysis of birth cohorts in South Asia: Way forward for Pakistan

Ahmed Waqas[1]*, Shamsa Zafar[2], Deborah A. Lawlor[3,4], John Wright[5], Assad Hafeez[6], Ikhlaq Ahmad[1,6], Siham Sikander[1,6], Atif Rahman[7]

1 Human Development Research Foundation, Markaz, Islamabad, Pakistan, 2 Department of Gynaecology and Obstetrics, Fazaia Medical College, Islamabad, Pakistan, 3 Medical Research Council Integrative Epidemiology Unit at the University of Bristol, University of Bristol, Bristol, United Kingdom, 4 Population Health Science, Bristol Medical School, Bristol University, Bristol, United Kingdom, 5 Bradford Institute for Health Research, Bradford Teaching Hospitals NHS Foundation Trust, Bradford, United Kingdom, 6 Health Services Academy, Chak Shahzad, Islamabad, Pakistan, 7 Institute of Psychology, Health and Society, University of Liverpool, Liverpool, United Kingdom

* ahmedwaqas1990@hotmail.com

**Data Availability Statement:** All relevant data are within the paper and its Supporting Information files.

**Funding:** This study has not received any direct funding. D.A.L works in a Unit that is supported by

## Abstract

The present study aims to: a) systematically map the of birth cohort studies from the South Asian region b) examine the major research foci and landmark contributions from these cohorts using reproducible scientometric techniques and c) offer recommendations on establishing new birth cohorts in Pakistan, building upon the strengths, weaknesses and gaps of previous cohorts. Bibliographic records for a total of 260 articles, published during through December 2018, were retrieved from the Web of Science (core database). All data were analysed using Microsoft Excel (2013), Web of Science platform and CiteSpace. A series of network analysis were then run for each time-period using the link reduction method and pathfinder network scaling. The co-cited articles were clustered into their homogeneous research clusters. The clusters were named using the Latent Semantic Indexing (LSI) method that utilized author keywords as source of names for these clusters. The scientometric analyses of original research output from these birth cohorts also paint a pessimistic landscape in Pakistan- where Pakistani sites for birth cohorts contributed only 31 publications; a majority of these utilized the MAL-ED birth cohort data. A majority of original studies were published from birth cohorts in India (156), Bangladesh (63), and Nepal (15). Out of these contributions, 31 studies reported data from multiple countries. The three major birth cohorts include prospective and multi-country MAL-ED birth cohort and The Pakistan Early Childhood Development Scale Up Trial, and a retrospective Maternal and infant nutrition intervention cohort. In addition to these, a few small-scale birth cohorts reported findings pertaining to neonatal sepsis, intrauterine growth retardation and its effects on linear growth of children and environmental enteropathy.

## Introduction

Longitudinal birth cohorts describe health and well-being throughout people's lives, investigating how social background, lifestyle and genetics, and other factors, work together to prevent,

the University of Bristol and UK Medical Research Council (MC_UU_00011/6). D.A.L's contribution to this work is also supported by the British Heart Foundation (AA/18/7/34219), European Research Council grant (669545) and a National Institute of Health Research Senior Investigator Award (NF-0616-10102). The views expressed in this publication are those of the authors and not necessarily any funding bodies. The specific roles of these authors are articulated in the 'author contributions' section. The funders had no role in study design, data collection and analysis, decision to publish, or preparation of the manuscript.

**Competing interests:** The authors of this paper have read the journal's policy and the authors have the following competing interests: D.A.L has received support from Roche Diagnostics and Metronic Ltd for research unrelated to this publication. All other authors report no conflict of interests to declare. There are no patents, products in development or marketed products to declare. This does not alter our adherence to PLOS ONE policies on sharing data and materials.

or cause ill-health [1–4]. They provide a test bed for understanding ways to support families and improve health. Importantly, they attract talented multi-disciplinary research teams, promote strong community engagement, and provide a catalyst for the translation of evidence into practice. The well-established birth cohorts in the high income western countries have made major contributions to science, policy and practice, and have provided important catalysts for social, biomedical and epidemiological research capacity building [5,6]. Moreover, birth cohorts associated with bio-banks have considerably advanced our understanding of the influence of genetics and epigenetics on disease burden [5,7].

South Asian countries face an epidemic of non-communicable and communicable diseases, with some of the world's highest rates of metabolic, cardiac and common mental disorders globally [8]. While South Asian countries have made substantial strides in reducing poverty, its performance on meeting the health related development goals have been less impressive [9]. This region accounts for 36% of the world's poor, nearly half of undernourished children, and perinatal and infant mortality. According to the United Nations' report, the region still lags behind 10/22 indicators for which reliable data is available [9]. It has been considered off-track on development goals pertaining to poverty, areas antenatal care, maternal, infant and under-5 morbidity and mortality [9]. Moreover, public health expenditures is extremely low at 1.3% of the GDP as compared to the world average of 6% [10]. The region also lags significantly behind in child education, due to poor quality of education and low public expenditure ranging from 1.7% (Sri Lanka) to 3.9% (India) of the GDP [10].

This double burden of poor health and psycho-social development in early life poses a grave threat to low and middle income countries [11,12]. Besides there is a strong evidence of early-life disadvantage contributing to higher rates of non-communicable diseases in later life [11]. For instance, low birth weight and poor maternal and foetal nutrition has been associated with increased coronary heart disease mortality, incidence of type-2 diabetes, hypertension, chronic kidney disease, COPD and poor neurodevelopment and mental health [13–19]. This situation is further worsened by an exponential growth in population and availability of scarce resources, which need to be channelled intelligently. This requires vital data at population level to aid in crucial policy and system level decision making. To address these challenges, longitudinal birth cohort studies provide large *Hadron Colliders* for health research, offering rich and varied resources for research spanning across life course and over generations.

Despite the high burden of morbidity and mortality in South Asia, very few longitudinal birth cohorts have been established here [20]. This is one of the culprits, leading to an inequity in global health research output and poor understanding of health determinants in the region. Therefore, establishment of longitudinal birth cohorts in the region, is often emphasized to help develop an understanding of health determinants. However, to date, there have been no evidence synthesis efforts pertaining to birth cohorts in South Asia. Thus, warranting this systematic mapping study in the South Asian context. The aims of this scientometric analyses are to a) systematically map birth cohort studies from the South Asian region and b) examine the major research foci and landmark contributions from these cohorts, using reproducible scientometric techniques.

We also offer recommendations on establishing a new birth cohort in Pakistan; a country that boasts a unique sociocultural setting in the region. Pakistan, officially the Islamic Republic of Pakistan, is a predominantly Muslim country. With a population exceeding 210 million people, it is the world's sixth most populous country (Pakistan Bureau of Statistics, 2017). It boasts a rich cultural and social diversity representing minorities from Hinduism, Christianity, Sikhism and Buddhism. Major ethnic groups include the Punjabis, Sindhis, Pashtuns and Baluchis as well as many other minority groups. Pakistan also houses one of the largest Afghan refugee population. In addition to its ethnic diversity, the nation of Pakistan is also rapidly

urbanizing- and has long suffered from political instability, wars and terrorism in the region. Pakistan also boasts a population growth rate of 2.40%- where children and adolescents make up over 35% of the population [21]. According to the World Health Organization, Pakistan also performs poorly on several socioeconomic and health indicators. Around 21% of the Pakistani population lives below the poverty line with poor access to healthcare. It also has one of the highest indicators of maternal and child mortality [21]. In 2018, Pakistan had a neonatal mortality rate of 42 (per 1000 live births), under five mortality rate of 69.3 and maternal mortality ratio of 140 per 100,000 live births [21].

## Methods

### Scientometric analysis: An introduction

This study was conducted as per the principles of knowledge mapping and co-citation analyses outlined by Chen et al., [22,23]. According to the theory of co-citation analyses, two studies (A & B) are said to be in a co-citation relationship when they are cited together by one study C [24]. The use of this theory has gained a recent momentum with the introduction of new analytical platforms such as the CiteSpace, Gephi and VOS viewer [22], which allow visualization of bibliographic data and their collaborative links.

In a broader context, Hess defined scientometrics as, the"quantitative study of science, communication in science, and science policy", helping to evaluate the impact of journals, scientists and institutes on the development and innovation of a scientific field" [25]. The main aim of a scientometric analysis is to evaluate research trends in a domain or discipline, to allow the mapping of new discoveries, landmark studies, and institutional (academic and funding) stakeholders [26,27]. These markers of scholarly activity together with infrastructural developments in a domain, provide important tools for institutions, research and funding agencies to identify areas where more research and funding is required [22,26]. Scientometric studies utilize several different analytical approaches, for instance, simplistic citation analyses reporting characteristics of top cited publications in a domain and by using more advanced techniques of network analysis to delineate co-citation relationships between important or top cited research studies and their cite references [26,28]. The scientometric practices are therefore, different to systematic reviews of literature, which aim to synthesize evidence and rate quality of literature pertaining to a focused research questions, such as the efficacy of a particular intervention [29].

### Academic database search

In December, 2018, we conducted an electronic search of Web of Science core databases with search terms, "((Birth-cohort) AND (Pakistan* OR India* OR Bangladesh* OR Afghanistan* OR Nepal* OR Bhutan* OR Maldives))"[30]. Region specific filters were applied to yield search results from the South Asian countries. And bibliographic records (including titles, author details, abstracts, characteristics of journal and citing references) for a total of 260 articles, published during through December 2018, were retrieved (Dataset 1). For the purpose of scientometric analyses, only Web of Science (core databases) were searched. This database records citing references of indexed studies, necessary for study of co-citation relationships in literature [22,23]. The performance of the keywords for database searches was assessed against several criteria, most importantly, by its ability to retrieve bibliographic records of South Asian birth cohorts identified by previous systematic reviews and indexing websites [31,32]. Our search strategy covered all the birth cohorts identified at a repository of birth cohort index (birthcohort.net), and thus, was judged as satisfactory [31,32].

As a precautionary measure and in response to a reviewer comment, a broader search strategy was performed using following keywords: TS = (Birth AND cohort) AND CU = (Pakistan* OR India* OR Bangladesh* OR Afghanistan* OR Nepal* OR Bhutan* OR Maldives)"[30]. This yielded a total of 982 titles and abstracts, which were carefully screened manually to see if they identified any additional birth cohorts in comparison with our primary planned search. After screening, we identified 214 relevant studies. Major reasons for exclusion of studies were: wrong study design (e.g. randomised trial, retrospective studies, case-control studies and cross-sectional studies (n = 710), populations other than mother infant-dyads (n = 32), postnatal recruitment (n = 14), wrong publication type (e.g. reviews (n = 8), countries other than South Asia (n = 5) and duplicate studies (n = 1). Notably, this broader search did not identify any additional South Asian birth cohorts in comparison to our main (initial planned) search.

## Scientometric analysis

The scientometric analysis and knowledge mapping was conducted with the software Citespace (v5.0 R2, Drexel University, Pennsylvania, USA). It is a Java-based, user friendly software that allows for knowledge mapping by visualization of bibliographic data [33–38]. The knowledge mapping in present study was predominantly based on the theory of co-citation analysis which considers a significant relationship between two or more articles when they are cited together in another publication [22,23]. For the purpose of visualization, the publication records were "sliced" into three time periods: 2000–2010; 2011–2015 and 2016–2018. Each time period was represented by a maximum of 50 top cited articles per year. The term sources selected were titles, abstracts, author keywords and keywords plus while nodes were characterized by cited references to allow for co-citation analysis [22,23]. The link strengths were determined with time slices using the Cosine method.

Articles were presented as nodes and link as edges. A series of network analysis were then run for each time-slice using the link reduction method and pathfinder network scaling [22,23]. The co-cited articles were clustered into their homogeneous research clusters. The clusters were named using the Latent Semantic Indexing (LSI) method, Log-likelihood ratio (LLR) and TF*IDF method. These methods utilized author keywords as source of names for these clusters [22,23]. These network analysis were then visualized to identify key results: a) key publications and significant entities controlling resources in their collaborative networks or cluster (represented with a centrality value > 0.1) b) landmark theories that act as a bridge between two different clusters, represented as purple nodes and c) articles with citations bursts representing hot topics of research in a specific time period [22,23]. Hot topics of research present short periods of citation bursts in a short period.

## Bibliometric analysis

For bibliometric analyses, the data set (n = 260) was screened to include original research publications published from the South Asian birth cohorts. Only original studies published from birth cohorts conducted in Pakistan, India, Bangladesh, Afghanistan, Nepal, Bhutan and Maldives were included. Out of the 260 full texts, 38 studies were excluded for major reason being study design other than birth cohort.

Then, data pertaining to characteristics of these studies were manually extracted from abstracts and entered in Microsoft Excel sheets. The data abstraction phase was performed by one reviewer (AW) who noted study characteristics including country, city/state, journal and year of publication, major research areas, funding and institutional affiliation. Moreover, during this data extraction exercise, unique cohorts were identified. Thereafter, several additional variables such as sample size, study design, primary hypotheses, time period, location and

variable measurements were extracted for this subset of cohorts. All data were analysed using Microsoft Excel (2013) and Web of Science analytics platform. A specialist programme (Stat-Planet MapMaker) was used to display the geographical spread of studies. Microsoft excel was used to calculate frequencies & percentages. Web of science analytics were used to identify top organizations, authors, funding agencies, and year-wise pattern of publication and citations.

## Results

### Descriptive and bibliometric analyses

**Research output.**   The scientometric analyses of original research output from these birth cohorts also paint a pessimistic landscape in Pakistan- where Pakistani sites for birth cohorts contributed only 31 publications, where a majority of these utilized the MAL-ED birth cohort data. Most original studies were published from birth cohorts in India (n = 156), Bangladesh (n = 63), and Nepal (n = 15). Out of these contributions, 31 studies reported data from multiple countries. The number of publications as well as received citations increased at a good rate since the year 2000 (Figs 1–3). The number of citations received by the publications rose from < 10 to over 750 from the year 2000 to 2018. While the number of articles rose from <5 to over 35 during the same period (Figs 1–3).

**Major birth cohorts in Pakistan.**   The three major birth cohorts include prospective and multi-country MAL-ED birth cohort and The Pakistan Early Childhood Development Scale Up Trial, and a retrospective Maternal and infant nutrition intervention cohort (13–15). The MAL-ED birth cohort included 274 participants from Pakistan for assessment of enteropathy and malnutrition in Pakistan (16). The PEDS trial conducted a 3-year follow-up 1,302 mother-child dyads who had participated in a nutrition trial (14). Similarly, the maternal and infant nutrition birth cohorts retrospectively followed up three nutrition intervention cohorts in the Sindh province (15). In addition to these, a few small-scale birth cohorts reported findings pertaining to neonatal sepsis (17)(18), intrauterine growth retardation and its effects on linear growth of children (19) and environmental enteropathy (20).

### Summary of pervious cohorts in South Asia (Table 1)

The well-established birth cohort studies in the South Asian region have helped elucidate underpinnings and trajectories of cardiovascular diseases, malnutrition and environmental enteropathies [4,39–43]. For instance, the contributions of the New Delhi Birth Cohort have helped elucidate the risk factors of cardiovascular diseases such as hypertension, atherosclerosis and myocardial infarction in India [42,44]. The Mysore Pathenon Cohort and the MAASTHI cohorts in India have contributed to our understanding of gestational diabetes and insulin sensitivity and their intergenerational effects such as adiposity in offspring [4,40]. And the multi-nationwide MAL-ED cohort has been crucial in understanding of enteric infections, diarrheal diseases and their effect on cognitive functions and use of antibiotics in countries including Pakistan, India, Bangladesh and Nepal [41,45]. The data from these cohorts has also allowed transcontinental comparisons yielding important findings such as the significant associations between earlier marriages and preterm births, low birth weight, poor schooling and metabolic syndrome among offspring [39].

### Scientometric & co-citation analyses

**Top organizations, funding bodies and journals.**   The top five organization contributing to research in this domain were Christian Medical College Hospital, India (n = 74); University of Southampton,UK (n = 34); University of Virginia, USA (n = 29); Aga Khan University,

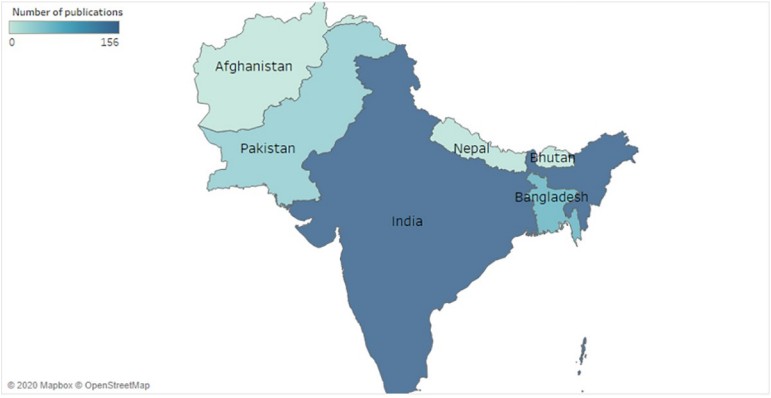

**Fig 1. Research productivity in South Asian region.** This figures the proportion of research output published from the South Asian countries.

Pakistan (n = 28), International Center for Diarrheal Disease Research, Bangladesh (n = 25). In addition to the number of publications, several more institutes were also identified as purple nodes (landmark contributors). These institutes included All India Institute of Medical Sciences, Emory University, World Health Organization and Johns Hopkins Bloomberg School of Public Health (Fig 4).

Top funding bodies were Medical Research Council, UK (n = 74), Wellcome Trust, UK (n = 43), Bill & Melinda Gates Foundation (n = 35), British Heart Foundation (n = 26) and the India Council of Medical Research (n = 24). The top 5 journals were American journal of Tropical Medicine and hygiene (n = 16), PlosOne (n = 14), Clinical infectious disease (n = 9), Indian Pediatrics (n = 9) and Vaccine (n = 8).

**Top keywords of birth cohort research in South Asia.** The top keywords in this domain fell into four main themes (Fig 5):

a. *Cardiovascular and metabolic health***:** This theme included keywords related to body mass index, obesity, blood pressure, insulin resistance, coronary heart disease, weight and birth weight.

b. *Nutrition***:** This theme spanned across following keywords: malnutrition and under-nutrition.

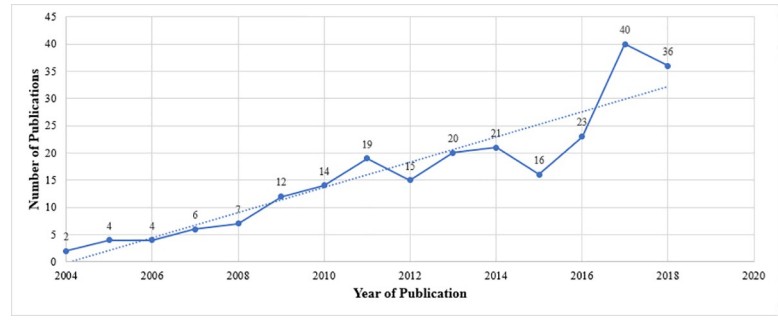

**Fig 2. Output of publications on birth cohort in South Asia region.** Number of publications pertaining to birth cohorts published in South Asia.

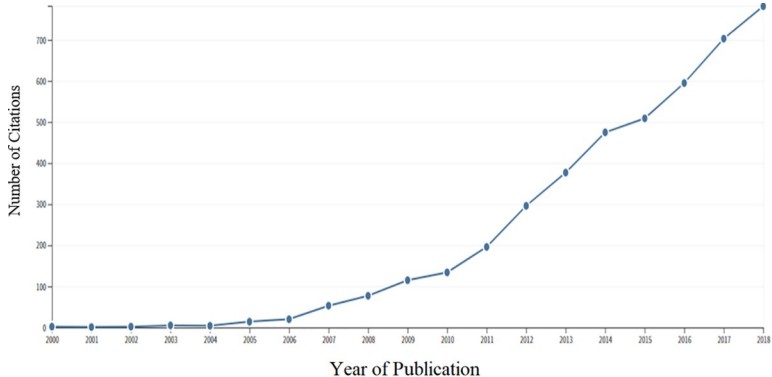

**Fig 3. Year-wise number of citations accrued by publications on birth cohorts published from South Asia.** Year-wise number of citations achieved by papers, pertaining to birth cohorts published in South Asia.

c. *Infections and vaccines*: It included keywords such as vaccines, diarrhea, infection, disease and mortality.

d. *Epidemiology*: It included keywords such as risk factor, prevalence, childhood, mortality, risk and young child burden.

## Foci of research & important studies from birth cohort research in South Asia

From the year 2005 to 2010, there were 546 nodes and 1447 edges (Fig 6). A total of 104 clusters were identified: 11 clusters representing a silhouette value of 1 and $\geq 10$ publications. The research focus in South Asian birth cohorts was mainly clustered in malnutrition and micronutrient supplementation of expectant mothers, coronary heart diseases, weight gain and adiposity, linear growth and stunting in the postnatal and adolescent period, allergic asthma and respiratory infections (RSV, rhinovirus), neonatal jaundice, and diarrheal infections, and effectiveness of rotavirus. There was some evidence of work on cervical cancer and still births.

During this period, six articles with centrality value > 0.1 were identified. Adair et al (2007) in their study of multiple birth cohort examined the inter-relationships of maternal factors during pregnancy, perinatal and early postnatal growth, and young adult anthropometric characteristics and resulting susceptibility to increased chronic disease risk [46]. Black & colleagues (2008), presented new insights pertaining to mortality and disability adjusted life years due to stunting, severe wasting and intra-uterine growth retardation [47]. They also reported that deficiencies of vitamin A and zinc were responsible for more 0.6 million deaths and sub-optimum breastfeeding was estimated to be responsible for 1·4 million child deaths worldwide. Gladstone et al, in the Vellore birth cohort found that children from urban slums were ill for approximately one fifth of infancy, mainly with respiratory and gastrointestinal illnesses [48]. Banerjee et al., (2006) documented significant genetic heterogeneity of rotaviruses in the community and the hospital, resembling a vaccine candidate strain caused disease in the community [49]. Building on the evidence from South Asian birth cohort studies, Baird et al (2005), conducted a meta-analysis that concluded that infants with higher BMI are at an increased of developing obesity in adolescence and adulthood [50]. Yajnik et al (2003) reported that Indian mothers but not fathers of heavier babies are at a higher risk of developing metabolic syndrome [51].

**Table 1. Description of key birth cohorts in South Asia.**

| Birth cohort | Study design, site, follow-ups, sample size | Aims & variables of interest |
|---|---|---|
| South Asian Birth Cohort (START)[78] | Prospective, Birth to age 3 years, 750 mother-infant dyads. | **Antenatal assessment:** sociodemography, maternal anthropometry, maternal glucose status, dietary assessment, physical activity, maternal depression, social support, acculturation and intimate partner violence, ultrasound examination of fetus. |
| | **Sites:** St. Johns Medical College Hospital, Bangalore; Snehalaya Hospital, Solur Village; Peel Ontario, Canada | **Delivery and newborn data:** crown-heel length, BMI, age and height till 18 months of age, skinfold thickness, bioelectrical impedance analysis, deuterium dilution analyses. |
| | | **Biological samples:** placental section, cord blood, blood samples for metabolic parameters and DNA. |
| | | **Postnatal data:** monthly weight and height, immunization status, breastfeeding practices and infant diet, sleep patterns using Brief infant sleep questionnaire, aspects of home environment and parent–child interaction, bonding, and child temperament. In India, the Bradley questionnaire will be used to capture information on parenting behavior under 6 domains (responsively, acceptance, organization, learning materials, parental involvement and variety of stimulation at home); child temperament using the Carey Temperment scales. |
| MAL-ED[45] | Prospective (2009–2013) Newborns followed till 24 months of age. | The Etiology, Risk Factors and Interactions of Enteric Infections and Malnutrition and the Consequences for Child Health and Development (MAL-ED). Enteropathogen infection contributes to: stunting, wasting, and micronutrient deficiencies; causes intestinal inflammation; cognitive impairments, and impaired responses to childhood vaccines. |
| | **Sites:** Dhaka, Bangladesh; Fortaleza, Brazil; Vellore, India; Bhaktapur, Nepal; Loreto, Peru; Naushahro Feroze, Pakistan; Venda, South Africa; and Haydom, Tanzania. | |
| | | **At birth:** Anthropometry, day and night blindness, and tobacco and alcohol use during pregnancy. |
| | | **Postnatal:** Surveillance of infectious diseases, general child health information, basic dietary intake, vaccination status, cognitive tests such as Bailey's infant development questionnaire, Infant Temperament Scale, MacArthur Adapted Communicative Development Inventory: Words and Gestures, maternal mood using self-reporting questionnaire. HOME Inventory |
| | | **Biological samples:** blood, urine, and monthly surveillance (non-diarrheal) and diarrheal stool samples to assess gut integrity, inflammation, prevalence of enteric pathogens, diarrheal illness surveillance, micronutrient levels, immunization and vaccine response. |
| The Maternal and Infant Nutrition Interventions in Matlab (MINIMat) cohort in Bangladesh[64] | Prospective 4436 mothers randomized; 2851 assessed at 4.5 years and 2307 at 12–14 years | **Interventions** Compliance food supplementation Compliance micronutrients (eDEM) |
| | Site: Matlab, Bangladesh | **Infant health:** gestational age (LMP, ultrasound), fetal growth (ultrasound), child anthropometry, skinfolds, body composition, child development Motor and cognitive development, language development, motor milestones, mother-child interaction, IQ, infections, immune function Morbidity, thymus size (ultrasound), |
| | | **Maternal health**: maternal anthropometry, reproductive history, previous pregnancies, outcomes, follow-up to next pregnancy. |
| | | **Diet:** food, diet, food security, breastfeeding |
| | | **Social conditions:** Household asset score Parents' education Parents' occupation Marital status Partner violence Depressive symptoms/distress, Home environment, |
| | | **Biological samples:**<br> **Biomarkers** haematology, micronutrient levels, oxidative stress, toxic exposure (urine) metabolic markers, blood pressure, salivary cortisol |
| **Andhra Pradesh Children and Parents Study (APCAPS)[79]** | Cross-sectional, prospective, part of a trial conducted in 1987–90 15 villages in Andhra Pradesh, India. Followups: 2003–2005 | Hyderabad Nutrition Trial (HNT) conducted in 1987–90 It explores the ddevelopmental origins of adult disease hypothesis'; undernutrition in early life plays a critical role in determining an individual's future risk of cardiovascular disease. |
| | | **Maternal and child variables:** Extensive data on socio-demographic, lifestyle, medical, anthropometric, physiological, vascular and body composition measures. |
| | | **Biological samples:** DNA, stored plasma, and assays of lipids and inflammatory markers on APCAPS participants are available. |

(*Continued*)

**Table 1.** (Continued)

| Birth cohort | Study design, site, follow-ups, sample size | Aims & variables of interest |
|---|---|---|
| **The Pakistan Early Childhood Development Scale Up Trial (PEDS)**[80] | Prospective 2009–2012 1302 mother-child dyads Enrolled at birth: follow ups at 2 and 4 years of age | The Pakistan Early Child Development Scale-up study assessed the longitudinal effectiveness of early nutrition and responsive stimulation interventions on growth and Development at 4 years of age. |
| | | **Variables:** |
| | | **Primary Outcome Measures:** Early Child Development: Language, Motor, Social-Emotional development Child Growth: Length/Height, Weight, Mid Arm Circumference, Head Circumference |
| | | **Secondary Outcome Measures:** Caregiving Mediators: Self-Reporting Questionnaire (SRQ)-20; Home Observation and Measurement of the Environment (HOME) Inventory: Mother/Child through Live observation; Care for Development; Knowledge and Practices Questionnaire (Maternal Report); Feeding Practices; Maternal report of infant and young child feeding practices; Morbidity; Anaemia Status |
| **IndEcho**[81] | Prospective 2016–2019 Recruiting 3,000 participants from New Delhi Birth Cohorts and Vellore Birth Cohort | Cohort study investigating birth size, childhood growth and young adult cardiovascular risk factors as predictors of midlife myocardial structure and function in South Asians. |
| | | **Variables:** Diet, physical activity, socioeconomic status, smoking, alcohol consumption, anthropometry, blood pressure. Biological samples: Glucose levels, insulin-fasting, cholesterol, triglycerides, HDL/LDL, urinary ACR, bioimpendance, hand grip, ECG, DXA, echocardiography, cIMT |
| **Mysore Birth Records Cohort in South India 1934–1966**[82] | Prospective First study: n = 1069 (199–2003) Second study: 521 (2013–2015) Third study: In process | First study examining relationship between birth size and adult CHD and risk factors Second study MYsore study of Natal effects on Ageing and Health—MYNAH* Third study: Lifecourse predictors of cognition in late life |
| | | **Variables:** Data wave for latest study includes variables as follows: Cognitive function, Geriatric mental state, chronic disease impairments, nutritional status, health behaviours and lifestyles, family living arrangements, economic status, social support and social networks: anthropometry, ECG, Rose angina questionnaire,32,33 blood pressure assessment, spirometry and a body composition analysis (bioimpedance) |
| | | **Biological samples:** blood tests for diabetes, insulin resistance, dyslipidaemia, anaemia, vitamin B12 and folate deficiency, hyper-homocysteinemia, renal impairment and thyroid disease; DNA sample for genetic assay of apoliprotein-E |

**Table 1.** (Continued)

| Birth cohort | Study design, site, follow-ups, sample size | Aims & variables of interest |
|---|---|---|
| **Mysore Parthenon Birth Cohort**[4] | Prospective 1997–98 recruitment 830 pregnant mothers | Coronary heart disease (CHD) and type 2 diabetes: long-term effects of maternal glucose tolerance and nutritional status on cardiovascular disease risk factors in the offspring. |
| | | Cardiovascular investigations were done at ages 5, 9.5 and 13.5 years in the children, and in the parents at pregnancy, 5-year and 9.5-year follow-ups. |
| | | **Variables:**<br>**During pregnancy:** 100-g, 3-h, oral glucose tolerance test<br>**Postnatal:** Serial anthropometry and body composition (bioimpedance), physiological and biochemical measures, dietary intake, nutritional status, physical activity measures, stress reactivity measures and cognitive function, and socio-demographic parameters for the offspring.<br>Data on anthropometry, cardiovascular risk factors and nutritional status are available for mothers during pregnancy.<br>**Biological samples:** Morning salivary cortisol, cortisol and cardiovascular responses to psychological stress, OGTT; HbA1c levels; fasting lipid concentrations; pulse, BP; plasma vitamin B12, folate, homocysteine; total and differential cell count; haemoglobin; blood grouping; hepatitis B status Actigraph accelerometer data. |
| MAASTHI: Maternal Antecedents of Adiposity and Studying the Transgenerational role of Hyperglycemia and Insulin[40] | Prospective Follow ups: 14 weeks, 1, 2,3,4 years | It explores maternal Antecedents of Adiposity and the tansgeneraional role of hyperglycemia and insulin |
| | | **Variables:**<br> Age, body mass index (BMI), family history of diabetes,gestational age, parity, past medical history, family history of hypertension and socio-economic status, developmental milestones<br> Maternal age, parity, BMI, weight-gain during pregnancy on fetal biometry measures, diet, gestational age, lifestyle factors, alcohol and tobacco use<br>**Maternal glucose in pregnancy**; skinfold thickness (adiposity) of offspring at one year<br>**Psychosocial environment:** social support and distress<br>**Biological samples** for insulin and genetic analyses, cord blood (n = 100) for c-peptide levels |
| Stress Responses in Adolescence and Vulnerability to Adult Non-communicable disease (SRAVANA) Study[83] | Prospective The study sample will be drawn from three well-established birth cohorts in India; the Parthenon cohort, Mysore (N = 550, age~20y), the SARAS KIDS prenatal intervention cohort, Mumbai (N = 300, age~10-12y) and the Pune Rural Intervention in Young Adults/ PRIYA cohort, Pune (N = 100, age~22y). | **Variables:**<br> 'Trier Social Stress Test (TSST)',<br>**Cardiometabolic parameters:**<br>Anthropometry, bioimpedance, hand-grip strength, blood pressure, fasting blood glucose, **Psychological parameters:**<br> PHQ-9, MINI, WISC-IV, Perceived stress scale, Stressful Life Events Scale, SDQ.<br>**Lifestyle indicators:**<br> Food frequency questionnaire, International physical activity questionnaire, smoking and alcohol intake, standard of living index<br>**Biological samples:**<br> Repeated measures of salivary cortisol and autonomic cardiovascular outcomes relative to the stressor will be assessed. Mechanistic studies including DNA methylation in gluco-corticoid receptor (NR3C1) and 11β-HSD2 gene loci and neuroimaging to measure Regional cerebral volumes in a subsample. |
| Performance of Rotavirus and Oral Polio Vaccines in Developing Countries" (PROVIDE) Study[65] | Prospective, RCT N = 700 | Efficacy of a 2-dose RotarixÒ oral rotavirus vaccine (given at 10 and 17 weeks of age) to prevent rotavirus diarrhea in the first year of life and OPV efficacy when a single inactivated polio vaccine (IPV) dose replaced the fourth dose of trivalent OPV (tOPV). The secondary objective was to determine whether EE, measured by lactulose/mannitol testing, was associated with reduced efficacy of oral vaccines for polio and rotavirus among infants |

(*Continued*)

**Table 1.** (Continued)

| Birth cohort | Study design, site, follow-ups, sample size | Aims & variables of interest |
|---|---|---|
| Vellore birth cohort study (1969–1973)[84] | Prospective n = 2218 (final dataset) Birth (1969–73) Follow-ups: Childhood and Adolescence (1977–80 and 1982–88) Re-trace the cohort in 1998 to 2002, then aged 26–32 years | The original study (1969–73) had five main objectives: (i) to study the relationship of birth weight and gestational age to infant mortality and the incidence of congenital defects; (ii) to study maternal blood pressure before and during pregnancy and the incidence of toxaemia; (iii) to assess the effects of parental consanguinity on reproductive outcomes; (iv) to examine the impact of family planning programmes on fertility; and (v) to estimate rates of fetal loss, and neonatal, infant and early childhood mortality.<br><br>The subsequent follow-up studies focused on the effects of prenatal factors, birthweight and gestational age on physical growth and development and mortality during childhood and adolescence.<br><br>For the follow-up in young adulthood (1998–2002), the main objective was to study glucose tolerance, and included measurements like insulin resistance and insulin secretion, and a range of cardiovascular risk factors (body composition, blood pressure and plasma lipid concentrations) in relation to parental size, neonatal size and childhood growth |
| Vellore cohort [85] | Prospective n = 497 | It explored the prenatal and postnatal risk factors for morbidity and growth in a birth cohort in southern India. |
| | | **Variables:**<br>Socioeconomic status, healthcare seeking practices, anthrompometry, morbidities such as GIT, respiratory, undifferentiated fever, skin infections and non-infectious morbidity |
| Maternal and infant nutrition intervention cohort [86] | Retrospective n = 1818 Urban and rural sites in Sindh, Pakistan: Karachi, Kotdiji, | This project followed three cohorts of children who received nutrition interventions in the first 1000 days of life. In the first cohort, maternal micronutrient supplementation was administered. The second cohort received micronutrient supplementation as newborns. The third cohort received complementary feeding strategies to support micronutrient status and child growth. |
| | | Now 4–9 years later, the children enrolled in these three interventions will be followed up and assessed on growth, developmental outcomes, and school performance. |
| | | **Variables:**<br>**Child outcomes:** morbidity, injuries and health (e.g. infection and hospitalization history), physical growth, body mass index and metabolic rate, nutrition and micronutrient status and development. |
| | | **Environment:** food security, physical environment and care-giving environment (i.e. learning and language stimulation). |
| | | **Maternal:** emotional well-being and reasoning ability. |
| | | **Blood sampling,** urine collections for iodine status, BIA and indirect calorimetry. |
| | | **Tools employed:** WISC IV; NEPSY; HOME Inventory; Wechsler Preschool and Primary Scale of Intelligence (WPPSI III); Raven's Coloured Progressive Matrices (RCPM). Strengths and Difficulties Questionnaire (SDQ); BOT-2 short form; SRQ-20 |

From the year 2011 to 2015 (Figs 7 & 8), there were 1656 nodes and 3704 edges and 461 recognized clusters. Out of these 461 clusters, only 16 clusters comprised ≥ 10 publications and a silhouette value of 1. Novel research foci emerged in addition to those identified in the previous time slice. This included bone mineral density, environmental enteropathy, respiratory tract infections and cognitive development of adolescents. However, a large portion of research remained concentrated in cardiometabolic health including the study of insulin sensitivity/diabetes/weight gain/body mass index/metabolic syndrome, arterial stiffness and hypertension. The work on cognitive development and cancers was only reported in a small number of publications.

During this period, eight research articles yielded a centrality value > 0.1, highlighting their significance in these collaborative networks. The most important publication was Adair et al's

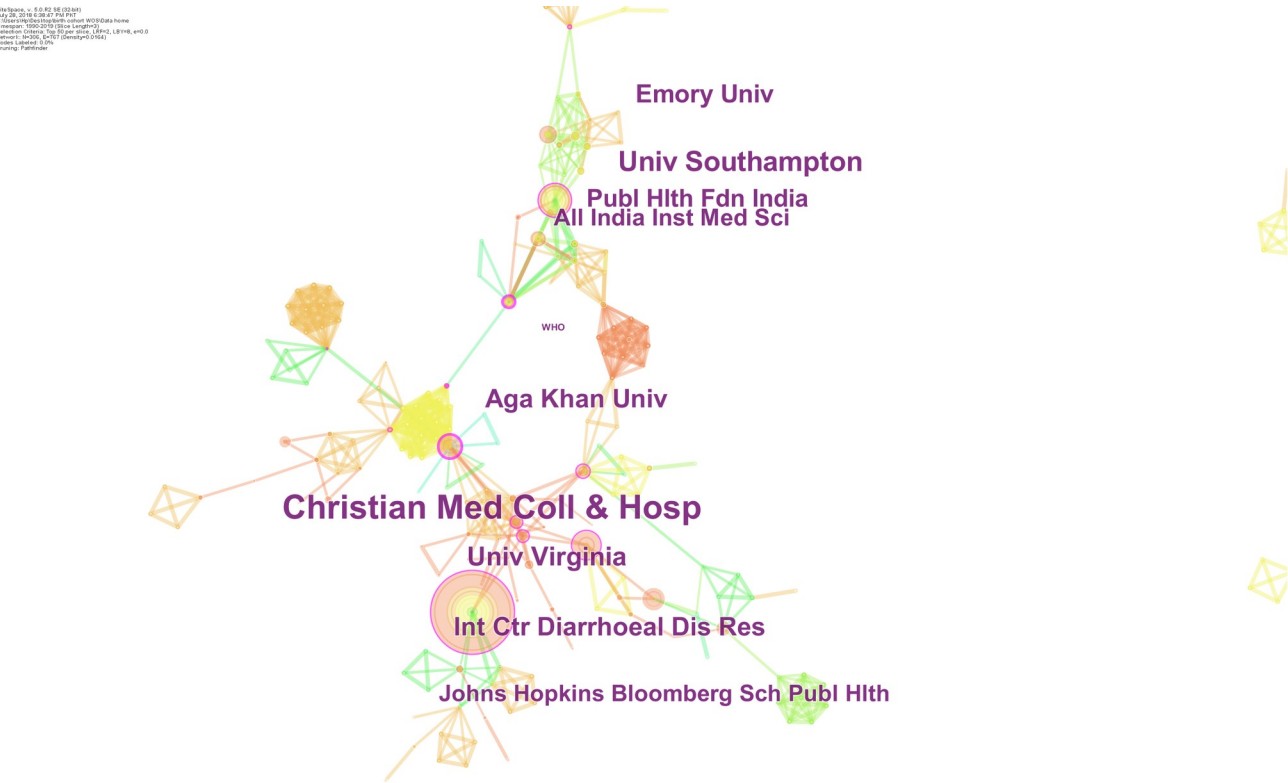

**Fig 4. Institutional collaborations on birth cohort research in South Asia.** Collaborative network of institutes in South Asia, working on birth cohort related research. Institutions are presented as nodes while the lines between them represent collaborative link structure. The circumference of the concentric ring is proportional to the number of citations accrued by the organization. Institutions presented as purple rings are important entities with landmark research profiles such as the International Centre of Diarrhoeal Disease Research.

analyses of multiple birth cohorts (2013), revealing a significant association between linear growth and relative weight gain during early life with adult health and human capital [46]. Araujo et al.'s (2006) work on obesity and breastfeeding "Pelotas 1993 birth cohort study", and Kaur et al's (2008) work influenced research in Indian birth cohorts [52,53]. Research in this period was significantly influenced by Jean Drèze & Amartya Sen's book delineating the deep divisions in Indian society and failure of public resources to enhance lives of the general public [54]. Nutrition intervention carried out by the Institute of Nutrition of Central America and Panama and the subsequent follow-ups was frequently analysed along with the data from New Delhi birth cohort. In 2010, Martorel & colleagues' leveraged the data from Panama and Guatemala highlighted the significant improvements in adult human capital and economic productivity resulting from the nutrition intervention given during early life [55]. Adair et al's (2007) on growth trajectories was a significant entity during this period as well. Lastly, the work conducted on pentavalent rotavirus vaccine against severe gastroenteritis influenced a lot of research in the MAL-ED associated countries [56].

From the year 2016–2018 (including late 2015), there were 609 nodes and 208 edges (Fig 9). The research was mainly focused on paediatric morbidities such as impaired growth, environmental enteropathy and intestinal infections caused by pathogens such as e-coli and parasites, systemic inflammation, and testing and up-scaling of rota virus vaccines mainly in South India and Bangladeshi health systems, behaviour change in Uttar Pradesh, exposure to chemicals and infant cognitive development. Among maternal morbidities, work was also conducted on

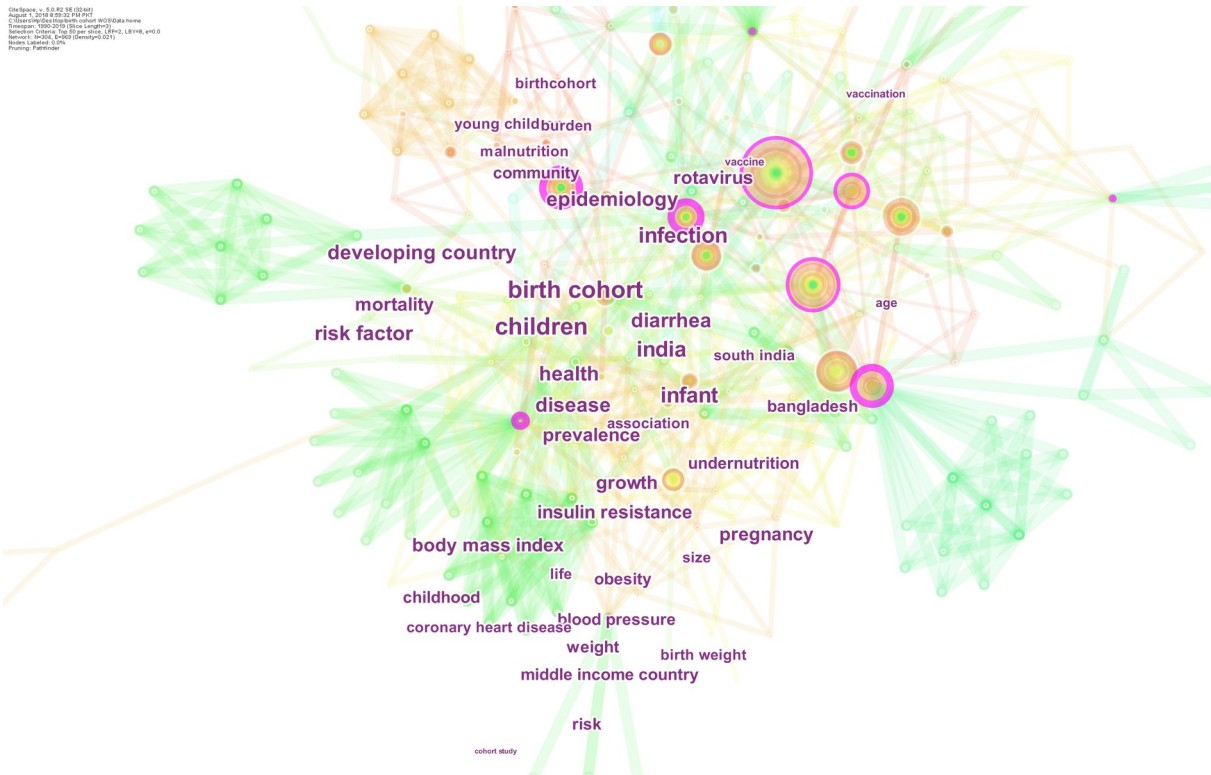

**Fig 5. Top keywords mentioned in birth cohort related publications in South Asia.** This figure presents top keywords used in birth cohort research in South Asia. Several important keywords attracting ground-breaking research are shown as purple rings, including infection, Bangladesh, Rotavirus, epidemiology and disease.

gestational diabetes, and breast-feeding patterns along with minor contributions to oropharyngeal cancer and genetics in the region (Fig 4).

During this period, no publication yielded a significant between-ness centrality- Nevertheless, the most important publications according to number of citations received were following: Platts-mills et al, (2015) utilized the data from MAL-ED and reported a substantial heterogeneity in pathogens causing diarrhea, with important determinants including age, geography, season, rotavirus vaccine usage, and symptoms, and concluded that single pathogen based vaccines may be of limited use in the region [57]. Kotloff et al, (2013) utilized the same dataset and reported rotavirus, Cryptosporidium, enterotoxigenic Escherichia coli producing heat-stable toxin and Shigella to be the major culprits causing moderate to severe diarrhoea [45]. Further major studies (Acosta et al, 2014; Kosek et al, 2013; Kosek et al, 2014), presented novel fecal markers "neopterin [NEO], alpha-anti-trypsin [AAT], and myeloperoxidase" that predict linear growth deficits among children with intestinal inflammation; furthering the understanding of causal pathways from enteropathogens to environmental enteropathy [58–61].

## Discussion

### Summary

Despite the striking burden of maternal and child morbidity and mortality concentrated in South Asia, very few birth cohorts have been established in the region. Since the year 2000, the number of publications from South Asian birth cohorts and their citation pattern have

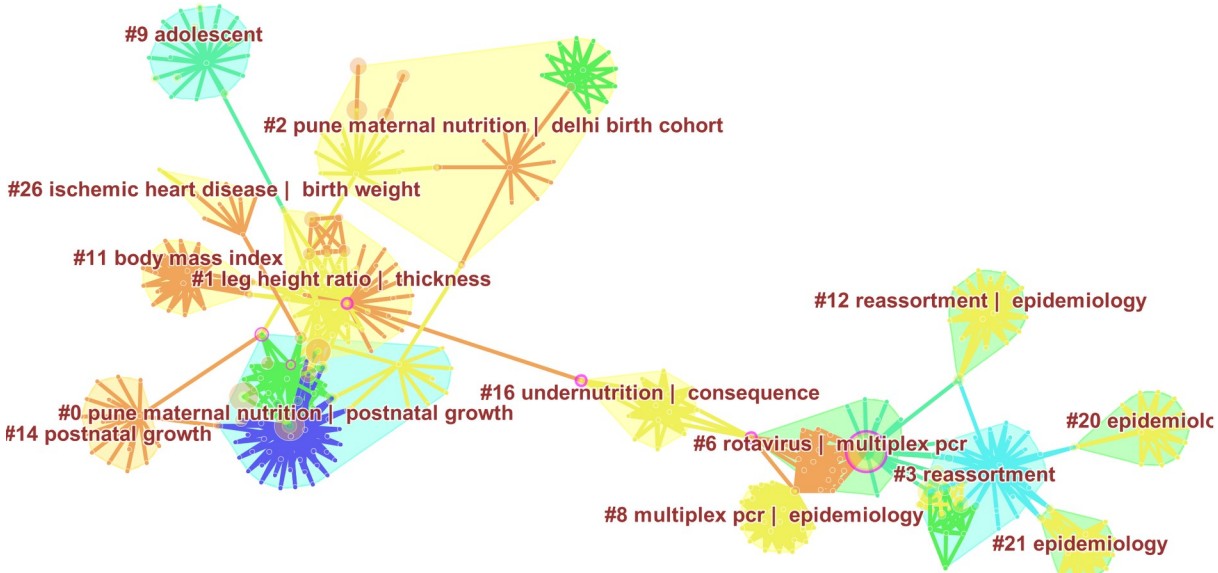

**Fig 6. Research Foci from 2005–2010.** This figure depicts major themes and foci of research output stemming from different birth cohorts established in South Asian Region, from 2005 to 2010. The shaded regions represent different clusters of research, labelled using indexed keywords. For instance, cluster # 1 in birth cohort research from 2005 to 2010 has been labelled as *leg height ratio and thickness*. Several nodes depicting number of citations as tree rings are shown to represent important entities. Purple coloured nodes represent landmark works that connect two clusters. Edges presenting relationship between two nodes are presented as lines. Different clusters are depicted with different colours.

followed an upward trend [62]. However, this finding is far from being encouraging when compared with the contributions from the developed nations. Moreover, the scientometric analyses of original research output from these birth cohorts also paint a pessimistic landscape in Pakistan- where Pakistani sites for birth cohorts contributed to only 31 publications, from South Asian [62].

## Need for establishing birth cohorts in Asia

Stakeholders running birth cohorts in developed nations have emphasized the need for region specific birth cohorts [63]. Findings from birth cohorts in Europe may not have the same relevance in other regions. In this context, Batty et al., opine that the nationals of developing nations nurture in environment that is unique on several levels [63]. For instance, presence of unique risk factors, varied composition of exposures as well as different confounding structures in countries at different stages of economic transition. Moreover, several European birth cohorts exploring associations between poor postnatal growth and health in adulthood, may have been more relevant in poorer nations stricken with malnutrition, and high rates of stunting and infectious illnesses. Countries at varying stages of economic transition may boast a different biopsychosocial environment owing to changes in lifestyle, dietary habits and political stability.

## Research gaps in South Asia

The present analyses revealed important research gaps in context of Asian birth cohorts. In addition to poor research output, most of the cohorts established in the South Asian region have research foci limited to cardiovascular diseases, linear growth of children malnutrition, enteric infections and environmental enteropathies [4,41,42,45,64,65]. And the study of their

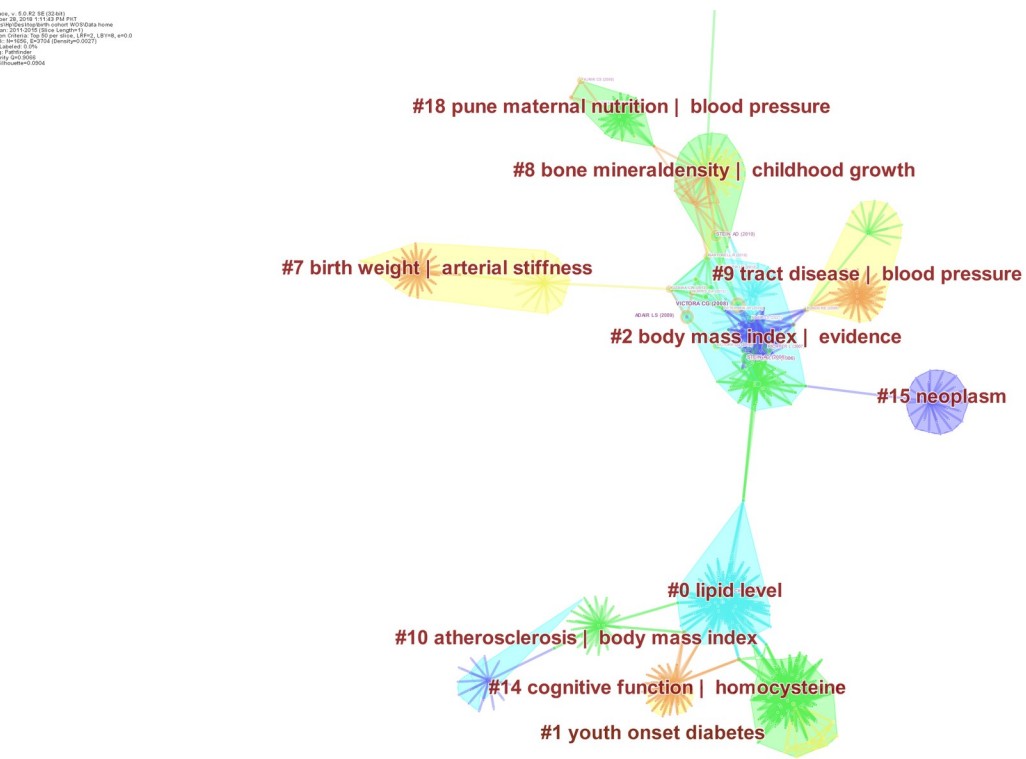

**Fig 7. Research foci from 2011-2015a.** This figure depicts major themes and foci of research output stemming from different birth cohorts established in South Asian Region, from 2011 to 2015. The shaded regions represent different clusters of research, labelled using indexed keywords. For instance, cluster # 0 is labelled as *lipid levels*. Several nodes depicting number of citations as tree rings are shown to represent important entities. Purple coloured nodes represent landmark works that connect two clusters. Edges presenting relationship between two nodes are presented as lines. Different clusters are depicted with different colours.

social determinant has either been completely ignored or seldom touched upon. In contrast, the contributions of landmark birth cohorts in high income countries have adopted a more holistic approach- with the study of psychosocial trajectories being an important part. This has led to several important discoveries linking the inutero and prenatal biopsychosocial environment with the cognitive development of offspring beyond the perinatal period [3,66–72]. In addition, the well-established biobanks storing biological specimens have helped in linking biomarkers of several diseases, and also their availability for future testing as the biological testing in proteomics, genetics and metabolomics further progresses.

## Way forward & priorities in Pakistan

- *Maternal, reproductive and infant health and development*: Collect detailed information on factors affecting maternal and infant health and development, to identify approaches to improve these key outcomes for future health and productivity and inform intervention studies.

- *Non-communicable diseases*: Identify early-life causes of, and predictors for, type 2 diabetes, cardiovascular disease and common mental disorder in Pakistan and inform novel interventions to prevent or address these.

**Fig 8. Research foci from 2011-2015b.** This figure depicts major themes and foci of research output stemming from different birth cohorts established in South Asian Region, from 2011 to 2015. The shaded regions represent different clusters of research, labelled using indexed keywords. For instance, cluster # 4 represents research on time series analysis. Several nodes depicting number of citations as tree rings are shown to represent important entities. Purple coloured nodes represent landmark works that connect two clusters. Edges presenting relationship between two nodes are presented as lines. Different clusters are depicted with different colours.

- *Health systems*: Allow systems level analysis of data and act as a stimulus and test bed for strengthening health systems building blocks, including staffing, information and service delivery.

- *Poverty*: Poverty is a key determinant of health, and poor health of a family member can lead to catastrophic poverty[73]. It should allow in-depth study of:

  i.   Interplay between health and poverty, and the extent to which there is a vicious cycle between the two across generations

  ii.  The extent to which poverty-reduction schemes, such as conditional cash transfer, both reduce poverty and improve health and free access to health, on long-term health goals.

  iii. Understanding the barriers to and facilitators of implementing poverty-reduction schemes by national and local governments, and access, uptake and continued participation in the schemes by different families.

  iv.  Understanding the relationship of poverty and poverty-reduction schemes to resilience and the ability to make health change and subsequent impacts on physiology, psychology and health.

**Fig 9. Research Foci from 2016–2018.** This figure depicts major themes and foci of research output stemming from different birth cohorts established in South Asian Region, from 2015 to 2018. The shaded regions represent different clusters of research, labelled using indexed keywords. For instance, cluster # 13 represents research on *environmental enteropathy*. Several nodes depicting number of citations as tree rings are shown to represent important entities. Purple coloured nodes represent landmark works that connect two clusters. Edges presenting relationship between two nodes are presented as lines. Different clusters are depicted with different colours.

- *Environment*: Pakistan is projected to be one of the most-affected country due to effects of climate change [74–76]. Over-crowding and water shortage are likely to be major challenges in the coming decade. It should help understand the impact of environment on health and solutions to reduce the impact, such as access to clean energy and water.

- *Genetics*: Pakistan has one of the highest rates of consanguineous marriages [77]. The high prevalence of human knockouts in the population is potentially highly informative about human health and development of innovative treatments.

- *Interdisciplinary collaborations*: Allow interdisciplinary collaborations within Pakistani and international academic communities. Develop the biobank and measurement variables keeping cutting edge technologies such as new techniques in genomics, nanotechnology and data sciences.

- Major research output in South Asia has been limited to the context of two countries: India and Bangladesh; and a limited aforementioned research foci. Therefore, there is a need to establish a birth cohort that answers research questions in Pakistani context, builds on the strengths and weaknesses of previous cohorts, and further augments their scope by taking mental health, social development and psychosocial environment in focus. This would also address the need of establishment of biological banks storing biological specimens. Thus, aptly putting psychosocial into cutting edge biological research.

• *Longer follow-ups*: Future cohort should prospectively follow their study samples for longer periods, and possibly over generations to allow for study trajectories of physical growth, cognitive development, and emergence of psychiatric disorders, effect of socio-political evolution of society and their trans-generational effects.

## Supporting information

**S1 Dataset.**
(TXT)

## Author Contributions

**Conceptualization:** Ahmed Waqas, Shamsa Zafar, Deborah A. Lawlor, John Wright, Assad Hafeez, Ikhlaq Ahmad, Siham Sikander, Atif Rahman.

**Data curation:** Ahmed Waqas, Shamsa Zafar, Ikhlaq Ahmad.

**Formal analysis:** Ahmed Waqas, Siham Sikander.

**Methodology:** Shamsa Zafar.

**Supervision:** Deborah A. Lawlor, John Wright, Atif Rahman.

**Validation:** Assad Hafeez.

**Visualization:** Ikhlaq Ahmad.

**Writing – original draft:** Ahmed Waqas, Atif Rahman.

**Writing – review & editing:** Shamsa Zafar, Deborah A. Lawlor, John Wright, Assad Hafeez, Ikhlaq Ahmad, Atif Rahman.

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
