## [Decision Letter · Decision Letter 0]

29 Oct 2019

PONE-D-19-24798

A systematic mapping study of birth cohorts in South Asia: Way forward for Pakistan

PLOS ONE

Dear Dr. Waqas,

Thank you for submitting your manuscript to PLOS ONE. After careful consideration, we feel that it has merit but does not fully meet PLOS ONE’s publication criteria as it currently stands. Therefore, we invite you to submit a revised version of the manuscript that addresses the points raised during the review process.

Please fully address all helpful and useful comments from the reviewer, including more comprehensive and detailed searching, extraction, coding and analysis, clearly distinguish between descriptive summary and analytical findings, avoiding cognitive leap from descriptive to inferences, and justification on whether this is a narrative review, systematic review, meta-analysis, systematic map. In addition, please address: a) Screening and extraction processesTo ensure a systematic and objective process of identifying relevant literature, please provide details on the pre-specified eligibility criteria for screening with reasons for exclusion, pre-defined data extraction form, number of researchers for independent searching, and how to handle any discrepancies. b) Uniqueness of PakistanPlease elaborate the uniqueness of Pakistan in South Asian settings, including similarities and differences in terms of historical context, social, economic and political development, and hence their exposures and confounding structures.

We would appreciate receiving your revised manuscript by Dec 13 2019 11:59PM. To enhance the reproducibility of your results, we recommend that if applicable you deposit your laboratory protocols in protocols.io, where a protocol can be assigned its own identifier (DOI) such that it can be cited independently in the future. For instructions see: http://journals.plos.org/plosone/s/submission-guidelines#loc-laboratory-protocols

Please include the following items when submitting your revised manuscript:A rebuttal letter that responds to each point raised by the academic editor and reviewer(s). This letter should be uploaded as separate file and labeled 'Response to Reviewers'.A marked-up copy of your manuscript that highlights changes made to the original version. This file should be uploaded as separate file and labeled 'Revised Manuscript with Track Changes'.An unmarked version of your revised paper without tracked changes. This file should be uploaded as separate file and labeled 'Manuscript'.

We look forward to receiving your revised manuscript.

Kind regards,

Man Ki Kwok

Academic Editor

PLOS ONE

Journal Requirements:

Reviewers' comments:

Reviewer's Responses to Questions

**Comments to the Author**

1. Is the manuscript technically sound, and do the data support the conclusions?

Reviewer #1: Partly

2. Has the statistical analysis been performed appropriately and rigorously? 

Reviewer #1: N/A

3. Have the authors made all data underlying the findings in their manuscript fully available?

Reviewer #1: No

4. Is the manuscript presented in an intelligible fashion and written in standard English?

Reviewer #1: Yes

5. Review Comments to the Author

Reviewer #1: To be considered a systematic map the authors need to provide:

- a rationale for searching only one database (minimum is usually two)

- to include the full search string for this search in a database

- full eligibility criteria

- a flow diagram (check PRISMA).

The systematic map findings are quite dense and hard to follow. It is not possible to ascertain from the methods how the studies were were coded lead to that analysis. They also veer from purely descriptive to reporting the potential contribution of studies to understand the burden of disease in the region. The latter appear to 'slip into the findings' here and there, leading the reader to make inferences, on studies which have not been critically appraised.

The review authors need to decide whether this is a literature review, systematic map or systematic map, and make appropropriate revisions accordingly before it can be sufficiently peer reviewed. The work will the have greater scientific value and contribution to make to the field.

6. PLOS authors have the option to publish the peer review history of their article (what does this mean?). If published, this will include your full peer review and any attached files.

Reviewer #1: No

---

## [Author Response · Author response to Decision Letter 0]

15 Nov 2019

Dear Dr. Man Ki Kwok,

 My co-authors & I are very grateful to you for an excellent feedback on the manuscript. We have revised our manuscript in line with your suggestions and provide line by line responses below. With these revisions, the quality of the manuscript has improved substantially.

I believe the major change in the revised manuscript is its shift from a systematic mapping study design to a scientometric and bibliometric study design. These methods are very common in library sciences and knowledge mapping fields but a rare occurrence in healthcare sciences. PloSOne has recently published several studies in this domain, including our recent paper on online hate:

Reference: Waqas A, Salminen J, Jung SG, Almerekhi H, Jansen BJ. Mapping online hate: A scientometric analysis on research trends and hotspots in research on online hate. PloS one. 2019 Sep 26;14(9):e0222194.

We have now described these methods in more detail and tried to draft it in a more reader friendly manner. We have also avoided cognitive leaps from descriptive to inferences and provided justification for the design of these study. Please, find detailed responses to each comment below. 

We are sure that this study would attract interdisciplinary leadership including scientists, policy makers and funders. It would also be a preamble to establishment of birth cohorts in Pakistan. We look forward to a favorable decision in due time. 

Best wishes,

Dr. Ahmed Waqas

Corresponding author

Editor comments

Comment 1

Please fully address all helpful and useful comments from the reviewer, including more comprehensive and detailed searching, extraction, coding and analysis, clearly distinguish between descriptive summary and analytical findings, avoiding cognitive leap from descriptive to inferences, and justification on whether this is a narrative review, systematic review, meta-analysis, systematic map. 

Response

Dear Dr. Man Ki Kwok, thank you for the excellent feedback. We believe these comments are very valid. We have revised our manuscript in line with these comments. This has raised the quality of the manuscript and improved the prospects of reproducibility. Please, find point to point responses below:

1) Comprehensive & detailed searching

For the purpose of this analysis, we only chose one database i.e. Web of Science (core databases). This is because unlike other databases such as scopus or pubmed; Web of Science has bibliographic details of indexed studies as well as the studies they are cited in. These bibliographic records thus, allow us to study co-citation relationships among different studies published in a domain. It follows the basic principle that if study A and B, both cite study C, they are said to have a co-citation relationship. 

We have therefore, added following information to reflect this:

Methods

Line 100-101: This study was conducted as per the principles of knowledge mapping and co-citation analyses outlined by Chen et al., [23,24]. 

Line 106-108: For the purpose of scientometric analyses, only Web of Science (core databases) were searched. This database records citing references of indexed studies, necessary for study of co-citation relationships in literature [23,24].

Comment 2

Extraction, coding and analysis

For this section, we have made the following changes:

a) In the methods section, we have clearly defined two phases of data analyses: bibliometric (manual data abstraction) and scientometric analyses (using citespace software).

b) Out of 260 full texts, 38 studies were excluded for reasons including regions other than South Asia (n= 21), other study designs (n= 11), duplicate texts (n= 4) and literature reviews or systematic reviews (n= 2). (Line 132 to 134).

c) For data abstraction procedures for bibliometric analyses, following information is added:

“Then, data pertaining to characteristics of these studies were manually extracted from abstracts and entered in Microsoft Excel sheets. The data abstraction phase was performed by one reviewer (AW) who noted study characteristics including country, city/state, journal and year of publication, major research areas, funding and institutional affiliation. Moreover, during this data extraction exercise, unique cohorts were identified. Thereafter, several additional variables such as sample size, study design, primary hypotheses, time period, location and variable measurements were extracted for this subset of cohorts”.

Comment 3

Clearly distinguish between descriptive summary and analytical findings, avoiding cognitive leap from descriptive to inferences

a) To distinguish between descriptive summary and analytical findings, we have restructured the results and discussion section. Results have now been divided into sections: i) Descriptive and bibliometric analyses ii) Scientometric & co-citation analyses 

This should distinguish between results obtained using manual abstraction of data and those using citespace software.

b) The section Descriptive and bibliometric analyses following subsections: Research output, Major birth cohorts in Pakistan and Summary of pervious cohorts in South Asia.

c) The section Scientometric & co-citation analyses has the following subsections: Top organizations, funding bodies and journals, Top keywords of Birth Cohort Research in South Asia, Foci of Research & important studies from Birth Cohort Research in South Asia.

d) We have also removed any points int his section that would better fit in discussion section. For instance, the following paragraph in results section has been deleted:

Co-citation analyses of the limited research contributions from birth cohort studies in South Asia reveal that these are clustered into a few specific clusters pertaining to physical health. And in contrast to the British cohorts, there has been negligible contributions in the area of psychological health of mothers and children. Moreover, the recent holistic drive in global health in delineating the psychoscocial determinants of diseases- thus, defining physical and psychiatric morbidities using both the biological and psychosocial approaches, is unseen in the region. 

Comment 4

Justification on whether this is a narrative review, systematic review, meta-analysis, systematic map

Response

We have now removed any mention of systematic mapping from the manuscript. The design of the study has now been correctly defined as scientometric analyses.

Comment 5

In addition, please address:

a) Screening and extraction processes

To ensure a systematic and objective process of identifying relevant literature, please provide details on the pre-specified eligibility criteria for screening with reasons for exclusion, pre-defined data extraction form, number of researchers for independent searching, and how to handle any discrepancies.

Response

We have now expanded the methods section on bibliometric analyses to give an account of identifying relevant literature, eligibility criteria for inclusion of studies and data extraction process. However, since this is not a systematic review, we did not follow PRISMA recommendations. These processes are also outlined in several publications in PloSOne:

1) Zongyi Y, Dongying C, Baifeng L. Global regulatory T-cell research from 2000 to 2015: a bibliometric analysis. PLoS One. 2016 Sep 9;11(9):e0162099.

2) Shen S, Cheng C, Yang J, Yang S. Visualized analysis of developing trends and hot topics in natural disaster research. PLoS one. 2018 Jan 19;13(1):e0191250.

3) Waqas A, Salminen J, Jung SG, Almerekhi H, Jansen BJ. Mapping online hate: A scientometric analysis on research trends and hotspots in research on online hate. PloS one. 2019;14(9).

In addition to changes mentioned above, the Bibliometric analysis in methods now includes following information:

For bibliometric analyses, the data set (n=260) was screened to include original research publications published from the South Asian birth cohorts. Only original studies published from birth cohorts conducted in Pakistan, India, Bangladesh, Afghanistan, Nepal, Bhutan and Maldives were included. Out of the 260 full texts, 38 studies were excluded for reasons including regions other than South Asia (n= 21), other study designs (n= 11), duplicate texts (n= 4) and literature reviews or systematic reviews (n= 2). Then, data pertaining to characteristics of these studies were manually extracted from abstracts and entered in Microsoft Excel sheets.

 Comment 6

Uniqueness of Pakistan

Please elaborate the uniqueness of Pakistan in South Asian settings, including similarities and differences in terms of historical context, social, economic and political development, and hence their exposures and confounding structures.

Response

A new paragraph on Pakistan’s unique sociocultural environment has been added in the revised manuscript:

We also offer recommendations on establishing a new birth cohort in Pakistan; a country that boasts a unique sociocultural setting in the region. Pakistan, officially the Islamic Republic of Pakistan, is a predominantly Muslim country. With a population exceeding 210 million people, it is the world’s sixth most populous country (Pakistan Bureau of Statistics, 2017). It boasts a rich cultural and social diversity representing minorities from Hinduism, Christianity, Sikhism and Buddhism. Major ethnic groups include the Punjabis, Sindhis, Pashtuns and Baluchis as well as many other minority groups. Pakistan also houses one of the largest Afghan refugee population. In addition to its ethnic diversity, the nation of Pakistan is also rapidly urbanizing- and has long suffered from political instability, wars and terrorism in the region. Pakistan also boasts a population growth rate of 2.40%- where children and adolescents make up over 35% of the population [21]. According to the World Health Organization, Pakistan also performs poorly on several socioeconomic and health indicators. Around 21% of the Pakistani population lives below the poverty line with poor access to healthcare. It also has one of the highest indicators of maternal and child mortality [21]. In 2018, Pakistan had a neonatal mortality rate of 42 (per 1000 live births), under five mortality rate of 69.3 and maternal mortality ratio of 140 per 100,000 live births [21]. 

Reviewer comments

Reviewer Comment #1: To be considered a systematic map the authors need to provide:

- a rationale for searching only one database (minimum is usually two)

- to include the full search string for this search in a database

- full eligibility criteria

- a flow diagram (check PRISMA).

Response

Dear Sir or Madam, thank you so much for such valuable feedback on the manuscript. Based on your feedback, I believe the major change in the revised manuscript is its shift from a systematic mapping study design to a scientometric and bibliometric study design. These methods are very common in library sciences and knowledge mapping fields but a rare occurrence in healthcare sciences. PloSOne has recently published several studies in this domain as noted above. Therefore, keeping in mind the study design of scientomric analyses, we have restructured the manuscript as detailed above. We have given a rationale for choosing only one database and the eligibility criteria. We have also given details of studies excluded in methods section with reasons. However, we did not provide a PRISMA flowchart which would have been more relevant for systematic or scoping reviews.

Comment 2

The systematic map findings are quite dense and hard to follow. It is not possible to ascertain from the methods how the studies were coded lead to that analysis. They also veer from purely descriptive to reporting the potential contribution of studies to understand the burden of disease in the region. The latter appear to 'slip into the findings' here and there, leading the reader to make inferences, on studies which have not been critically appraised.

Response

We have removed any points in this section that would better fit in discussion section. Moreover, we have also deleted sentences related to quality appraisals of studies that was not an objective of this study. To make the processes of data abstraction clearer, we have divided the methods and results section into two sections as noted above.

To avoid any cognitive leaps from purely descriptive to reporting the potential contribution of studies. For instance, the following paragraph in results section has been deleted:

Co-citation analyses of the limited research contributions from birth cohort studies in South Asia reveal that these are clustered into a few specific clusters pertaining to physical health. And in contrast to the British cohorts, there has been negligible contributions in the area of psychological health of mothers and children. Moreover, the recent holistic drive in global health in delineating the psychoscocial determinants of diseases- thus, defining physical and psychiatric morbidities using both the biological and psychosocial approaches, is unseen in the region. 

Comment 3

The review authors need to decide whether this is a literature review, systematic map or systematic map, and make appropriate revisions accordingly before it can be sufficiently peer reviewed. The work will the have greater scientific value and contribution to make to the field.

Response

We have now removed any mention of systematic mapping from the manuscript. The design of the study has now been correctly defined as scientometric analyses.

---

## [Decision Letter · Decision Letter 1]

3 Mar 2020

PONE-D-19-24798R1

A scientometric analysis of birth cohorts in South Asia: Way forward for Pakistan

PLOS ONE

Dear Dr. Waqas,

Thank you for submitting your manuscript to PLOS ONE. After careful consideration, we feel that it has merit but does not fully meet PLOS ONE’s publication criteria as it currently stands. Therefore, we invite you to submit a revised version of the manuscript that addresses the points raised during the review process.

Please fully address all helpful and useful comments from the reviewer as attached, including revising search results using broader terms, and justifying the importance and relevance of insights gained and results generated from this study especially for the wider international community.

We would appreciate receiving your revised manuscript by Apr 17 2020 11:59PM. To enhance the reproducibility of your results, we recommend that if applicable you deposit your laboratory protocols in protocols.io, where a protocol can be assigned its own identifier (DOI) such that it can be cited independently in the future. For instructions see: http://journals.plos.org/plosone/s/submission-guidelines#loc-laboratory-protocols

We look forward to receiving your revised manuscript.

Kind regards,

Man Ki Kwok

Academic Editor

PLOS ONE

Reviewers' comments:

Reviewer's Responses to Questions

**Comments to the Author**

1. If the authors have adequately addressed your comments raised in a previous round of review and you feel that this manuscript is now acceptable for publication, you may indicate that here to bypass the “Comments to the Author” section, enter your conflict of interest statement in the “Confidential to Editor” section, and submit your "Accept" recommendation.

Reviewer #2: All comments have been addressed

2. Is the manuscript technically sound, and do the data support the conclusions?

Reviewer #2: Partly

3. Has the statistical analysis been performed appropriately and rigorously? 

Reviewer #2: No

4. Have the authors made all data underlying the findings in their manuscript fully available?

Reviewer #2: Yes

5. Is the manuscript presented in an intelligible fashion and written in standard English?

Reviewer #2: Yes

6. Review Comments to the Author

Reviewer #2: General comments

Ahmed Waqas and colleagues conducted a study on bibliographic records retrieved from the Web of Science to assess systematically map of birth cohort studies from the South Asian region, examine the major research foci and landmark contributions, and then recommend establishing new birth cohorts in Pakistan.

260 articles, published during through December, 2018, were retrieved from the Web of Science core databases based on search string: “((Birth-cohort) AND (Pakistan* OR India* OR Bangladesh* OR Afghanistan* OR Nepal* OR Bhutan* OR Maldives))”.

They found that India, Bangladesh, and Nepal published majority of original studies in birth cohorts. Three major birth cohorts were the prospective and multi-country MAL-ED birth cohort, the Pakistan Early Childhood Development Scale Up Trial, and a retrospective Maternal and infant nutrition intervention cohort.

Although reviewers’ comments were carefully responded, there are some issues needed to be clarified.

Specific comments:

1. The search string was too strict. Only term “birth-cohort” was used to search on the Web of Science. When other terms (e.g. “birth cohort” or “birth cohorts”) were added into the search string, the number of articles was ~ three-fold increase (277 vs 882 - see the figure). This leads to a concern about the results of this study.

2. There is not much insight in the topic and the results of this study. Personally, I think if authors check their search string and correct their data, it is suitable for publishing in a local journal.

7. PLOS authors have the option to publish the peer review history of their article (what does this mean?). If published, this will include your full peer review and any attached files.

Reviewer #2: No

---

## [Author Response · Author response to Decision Letter 1]

17 Mar 2020

Detailed letter to reviewers has been provided as a word file.

---

## [Decision Letter · Decision Letter 2]

15 Apr 2020

PONE-D-19-24798R2

A scientometric analysis of birth cohorts in South Asia: Way forward for Pakistan

PLOS ONE

Dear Dr. Waqas,

Thank you for submitting your manuscript to PLOS ONE. After careful consideration, we feel that it has merit but does not fully meet PLOS ONE’s publication criteria as it currently stands. Therefore, we invite you to submit a revised version of the manuscript that addresses the points raised during the review process.

Please fully address all helpful comments on Figures from the reviewer.

We would appreciate receiving your revised manuscript by May 30 2020 11:59PM. To enhance the reproducibility of your results, we recommend that if applicable you deposit your laboratory protocols in protocols.io, where a protocol can be assigned its own identifier (DOI) such that it can be cited independently in the future. For instructions see: http://journals.plos.org/plosone/s/submission-guidelines#loc-laboratory-protocols

We look forward to receiving your revised manuscript.

Kind regards,

Man Ki Kwok

Academic Editor

PLOS ONE

Reviewers' comments:

Reviewer's Responses to Questions

**Comments to the Author**

1. If the authors have adequately addressed your comments raised in a previous round of review and you feel that this manuscript is now acceptable for publication, you may indicate that here to bypass the “Comments to the Author” section, enter your conflict of interest statement in the “Confidential to Editor” section, and submit your "Accept" recommendation.

Reviewer #2: All comments have been addressed

2. Is the manuscript technically sound, and do the data support the conclusions?

Reviewer #2: Yes

3. Has the statistical analysis been performed appropriately and rigorously? 

Reviewer #2: Yes

4. Have the authors made all data underlying the findings in their manuscript fully available?

Reviewer #2: No

5. Is the manuscript presented in an intelligible fashion and written in standard English?

Reviewer #2: Yes

6. Review Comments to the Author

Reviewer #2: Figure 1 needs to show name of each country in the map. It is better to use shades of a color to reflect the numbers of original publications rather than use different colors. Figure 2 & 3 do not have x-axis and y-axis labels. The figures also need to show values in bars and lines . Figure 4, 5, 6, & 7 are quite dense and hard to follow. Authors should provide more description in figures to make reader understand what information represented (e.g. edges, nodes, clusters, colors) and the key points they want to show. Line 249 mentions three main themes but the text lists 4 themes (a, b, c, d). It will be easier to follow if these themes are demonstrated in Figure 5. Some labels in Figure 7 are overlapped.

7. PLOS authors have the option to publish the peer review history of their article (what does this mean?). If published, this will include your full peer review and any attached files.

Reviewer #2: No

---

## [Author Response · Author response to Decision Letter 2]

12 May 2020

A detailed letter to reviewer has been provided as a supplementary file.

---

## [Decision Letter · Decision Letter 3]

10 Jun 2020

PONE-D-19-24798R3

A scientometric analysis of birth cohorts in South Asia: Way forward for Pakistan

PLOS ONE

Dear Dr. Waqas,

Thank you for submitting your manuscript to PLOS ONE. After careful consideration, we feel that it has merit but does not fully meet PLOS ONE’s publication criteria as it currently stands. Therefore, we invite you to submit a revised version of the manuscript that addresses the points raised during the review process.

Please fully address the edits on Figures suggested by the reviewer.

We look forward to receiving your revised manuscript.

Kind regards,

Man Ki Kwok

Academic Editor

PLOS ONE

Reviewers' comments:

Reviewer's Responses to Questions

**Comments to the Author**

1. If the authors have adequately addressed your comments raised in a previous round of review and you feel that this manuscript is now acceptable for publication, you may indicate that here to bypass the “Comments to the Author” section, enter your conflict of interest statement in the “Confidential to Editor” section, and submit your "Accept" recommendation.

Reviewer #2: All comments have been addressed

2. Is the manuscript technically sound, and do the data support the conclusions?

Reviewer #2: Yes

3. Has the statistical analysis been performed appropriately and rigorously? 

Reviewer #2: Yes

4. Have the authors made all data underlying the findings in their manuscript fully available?

Reviewer #2: Yes

5. Is the manuscript presented in an intelligible fashion and written in standard English?

Reviewer #2: Yes

6. Review Comments to the Author

Reviewer #2: Figure 1: Please check the colors of Bhutan and Afghanistan. It is fine to use different colors to demonstrate countries. However, a specific color pallet or different shades of one color are better to reflect the difference in numbers of publications among the countries.

Figure 2: Please be specific in Y-axis (i.e. Number of Publications). Show the number in each bar.

Figure 3: It is necessary to distinguish between rate/ frequency and number of citations. The text mentioned “number of citations” and the Y-axis shows “Frequency of citations” while the label indicates “The citation rates of citations”.

Figure 6,7,8,9: In each figure, please describe the information that the edge, node, clusters, and color indicate.

7. PLOS authors have the option to publish the peer review history of their article (what does this mean?). If published, this will include your full peer review and any attached files.

Reviewer #2: No

---

## [Author Response · Author response to Decision Letter 3]

14 Jun 2020

A detailed letter has been uploaded as a supplementary file.

---

## [Editor Report · Decision Letter 4]

16 Jun 2020

A scientometric analysis of birth cohorts in South Asia: Way forward for Pakistan

PONE-D-19-24798R4

Dear Dr. Waqas,

We’re pleased to inform you that your manuscript has been judged scientifically suitable for publication and will be formally accepted for publication once it meets all outstanding technical requirements.

Kind regards,

Man Ki Kwok

Academic Editor

PLOS ONE
---

## [Editor Report · Acceptance letter]

19 Jun 2020

PONE-D-19-24798R4 

A scientometric analysis of birth cohorts in South Asia: Way forward for Pakistan 

Dear Dr. Waqas:

I'm pleased to inform you that your manuscript has been deemed suitable for publication in PLOS ONE. Congratulations! Your manuscript is now with our production department. 

Kind regards, 

on behalf of

Dr. Man Ki Kwok 

Academic Editor

PLOS ONE